# A recombinant human immunoglobulin with coherent avidity to hepatitis B virus surface antigens of various viral genotypes and clinical mutants

Gi Uk Jeong[1], Byung-Yoon Ahn[1]*, Jaesung Jung[2], Hyunjin Kim[2], Tae-Hee Kim[2], Woohyun Kim[2], Ara Lee[2], Kyuhyun Lee[3], Jung-Hwan Kim[2]*

1 Department of Life Science, Korea University, Seoul, Korea, 2 Mogam Institute for Biomedical Research, Youngin, Korea, 3 Development Division PL Unit, GC Pharma Corp., Youngin, Korea

* ahnbyung@korea.ac.kr (BYA); lmv338@gmail.com (JHK)

**Data Availability Statement:** All relevant data are within the manuscript and Supporting Information files.

## Abstract

The hepatitis B virus (HBV) envelope is composed of a lipid bilayer and three glycoproteins, referred to as the large (L), middle (M), and small (S) hepatitis B virus surface antigens (HBsAg). S protein constitutes the major portion of the viral envelope and an even greater proportion of subviral particles (SVP) that circulate in the blood. Recombinant S proteins are currently used as a preventive vaccine, while plasma fractions isolated from vaccinated people, referred to as hepatitis B immune globulin (HBIG), are used for short-term prophylaxis. Here, we characterized a recombinant human IgG1 type anti-S antibody named Lenvervimab regarding its binding property to a variety of cloned S antigens. Immunochemical data showed an overall consistent avidity of the antibody to S antigens of most viral genotypes distributed worldwide. Further, antibody binding was not affected by the mutations in the antigenic 'a' determinant found in many clinical variants, including the immune escape mutant G145R. In addition, mutations in the S gene sequence that confer drug resistance to the viral polymerase did not interfere with the antibody binding. These results support for a preventive use of the antibody against HBV infection.

## Introduction

More than 250 million people worldwide carry hepatitis B virus (HBV) in a chronic state, which may develop into serious liver diseases, such as cirrhosis and liver cancer [1]. S protein, the smallest (226 amino acids) but most abundant of the three viral surface antigens (HBsAg), constitutes a major portion of virion particles and an even greater portion of subviral particles (SVP) in spherical and tubular forms, often in 100,000-fold excess amount as that of virion particles in the blood of infected individuals (Reviewed in [2]). Currently, vaccines made of S protein are used for preventive purposes (Reviewed in [3]), while the human plasma antibody fraction, called hepatitis B immunoglobulin (HBIG), is used for short-term prophylaxis for newborns of chronic carrier parents and for immunosuppressed patients [4]. Regarding cost

**Funding:** This study was supported by the Basic Science Research Program Grant (#2018050379) of the Korea National Research Foundation (to B. Ahn) and by the collaborative research program of the Mogam Institute for Biomedical Research. G.U. Jeong was supported by the BK21 Plus program of the Ministry of Education of the Republic of Korea.

**Competing interests:** The authors have declared that no competing interests exist.

and safety issues, however, a recombinant anti-S antibody could be a good alternative/substitute for HBIG [5].

Lenvervimab is a human IgG1-type recombinant monoclonal anti-S antibody produced from CHO cells stably transfected with a cloned human immunoglobulin gene [6]. The HBV neutralizing activity of Lenvervimab was previously addressed utilizing a chimpanzee animal model [7]. In the current work, to address its preventive utility against HBV infection we characterized the antigen binding property of Lenvervimab to S antigens of various genotypes and clinical variants. Our results indicate that the antibody binds with an overall consistent avidity to S antigens from most viral genotypes distributed worldwide and most clinical variants with mutations in the 'a' determinant of S antigens, a dominant antigenic region conserved in all HBV strains. Further, antibody binding was not grossly affected by mutations in the S gene region that overlap with drug resistance mutations found in the viral polymerase. These results support the potential utility of Lenvervimab as a preventive measure against HBV infection.

## Results

### Lenvervimab neutralizes HBV infectivity in culture

The HBV neutralizing activity of Lenvervimab had previously been demonstrated in a chimpanzee animal model [7]. In the current study, HBV neutralization was examined in cell culture. Human hepatoma HepG2 cells that stably express the cellular receptor protein for HBV sodium taurocholate cotransporter protein (NTCP) were infected with HBV (Genotype D, ayw) in the presence of 0.001~1 microgram/ml Lenvervimab. Viral expression of the surface antigen HBsAg and the core protein HBcAg was inhibited by ~70% at 1 microgram/ml of Lenvervimab or equivalent concentration of a commercial rabbit polyclonal anti-HBsAg antibody, indicating a partial neutralization (Fig 1).

### Lenvervimab binds only nondenatured forms of HBsAg

Earlier studies have investigated the specificity and avidity of anti-HBsAg antibodies to a variety of HBsAg expressed from cloned genes. Differences in the expression and antibody binding affinity of each clones, however, made it difficult to compare the quantity and binding affinity of individual clones by immunochemical methods [8, 9]. For a more reliable quantification of the antigens, we tagged a hemagglutinin (HA) epitope to the N-terminus of all S clones. Before examining a variety of S clones, we evaluated the strategy for detection of wild-type S proteins. In the first set of blots, an aliquot of culture supernatant was directly resolved in an agarose gel in nondenaturing conditions, following the protocol initially developed for virion particle gel assay [10]. Probing the capillary-transferred membrane with Lenvervimab revealed a broad band that migrated slightly faster than an equivalent sample from HBV1.3-transfected cells (Fig 2). A similar S band was detected by anti-HA antibody from an independent gel. In contrast, no signal was detected from a denaturing polyacrylamide gel blot by Lenvervimab, whereas anti-HA antibody detected a protein of 24 kDa and another one, which is slightly larger in size probably due to glycosylation. These results indicate that Lenvervimab binding requires an exclusively nondenaturing condition, whereas anti-HA antibody binds the HA-tagged S protein in both conditions. It is also evident that HA-tagged S protein is expressed in particle forms, much like the subviral particles (SVP) produced from HBV DNA-transfected cells as reported [11]. Although a fully enveloped virion could have also been produced from the replication competent HBV1.3 DNA, we could not separate or distinguish the two fractions in this experiment. Earlier studies showed that SVP in blood circulation is composed of some lipids and ~100 molecules of HBsAg, the majority of which are composed of S proteins

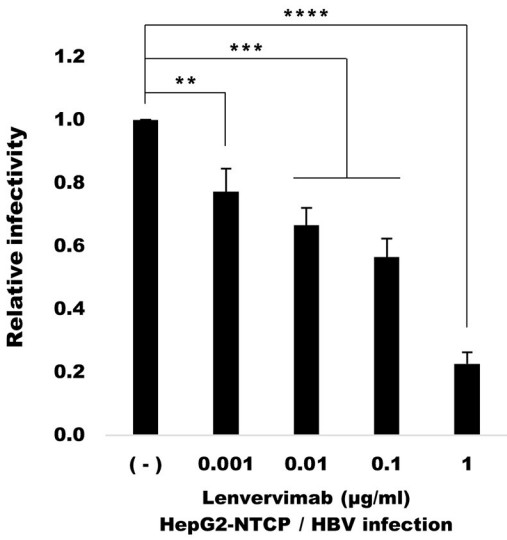
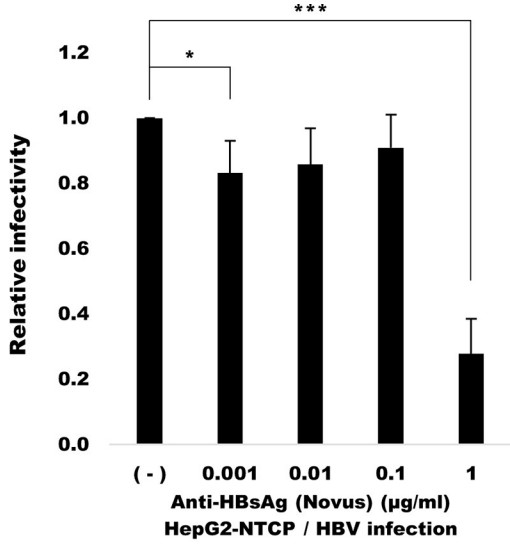

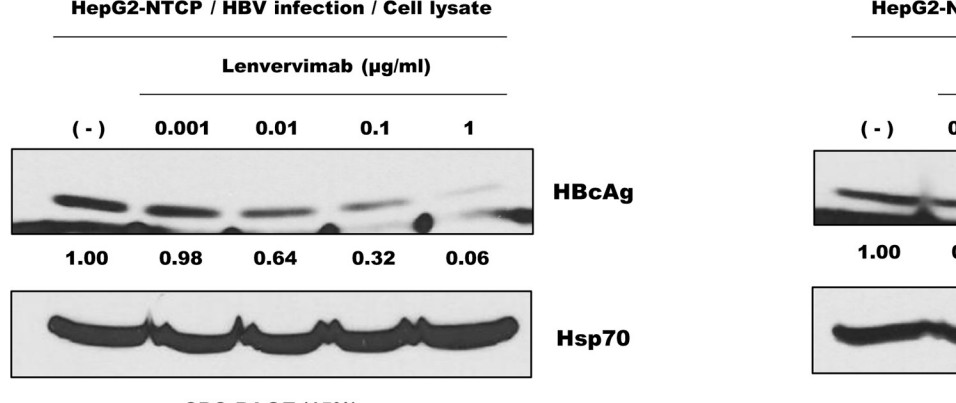
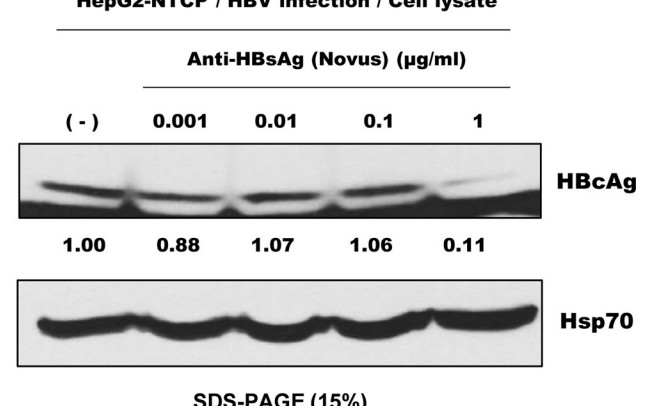

**Fig 1. Lenvervimab neutralizes HBV infection in culture.** (Upper) HepG2-NTCP cells were infected for 16 h with HBV at 500 Geq/cell in the presence of 0.001~1 microgram/ml of Lenvervimab (shown on the left) or a commercial rabbit polyclonal anti-HBsAg antibody (from Novus Biologicals) (shown on the right). HBsAg secreted into the culture media was measured after 5 days by ELISA. Bars indicate the average value and standard errors of triplicate experiments. P-values were obtained by utilizing the Excel software based on the two-tailed Student's t-test. P-values indicated are less than 0.05 (*), 0.01 (**), 0.001 (***) and 0.0001 (****), respectively. (Lower) Intracellular viral core protein (HBcAg) after 5 days was examined by immunoblotting with an anti-HBcAg antibody. Values shown below the blot are the intensity of HBcAg scanned by Image J software (NIH) and normalized with that of the loading control Hsp70.

[12, 13]. The quantity of S antigen produced in our transfection analysis ranged within ~100 ng/ml of culture supernatant.

## Lenvervimab binds to the S antigens of all viral genotypes

HBV distributed worldwide is classified into eight distinct (Genotypes A to H) and two tentative (Genotypes I and J) groups based on their genome sequence differences [14]. Some genotypes are found to be associated with geographic prevalence and clinical features of disease [15]. Although S gene sequence variations are also found between different genotypes, their

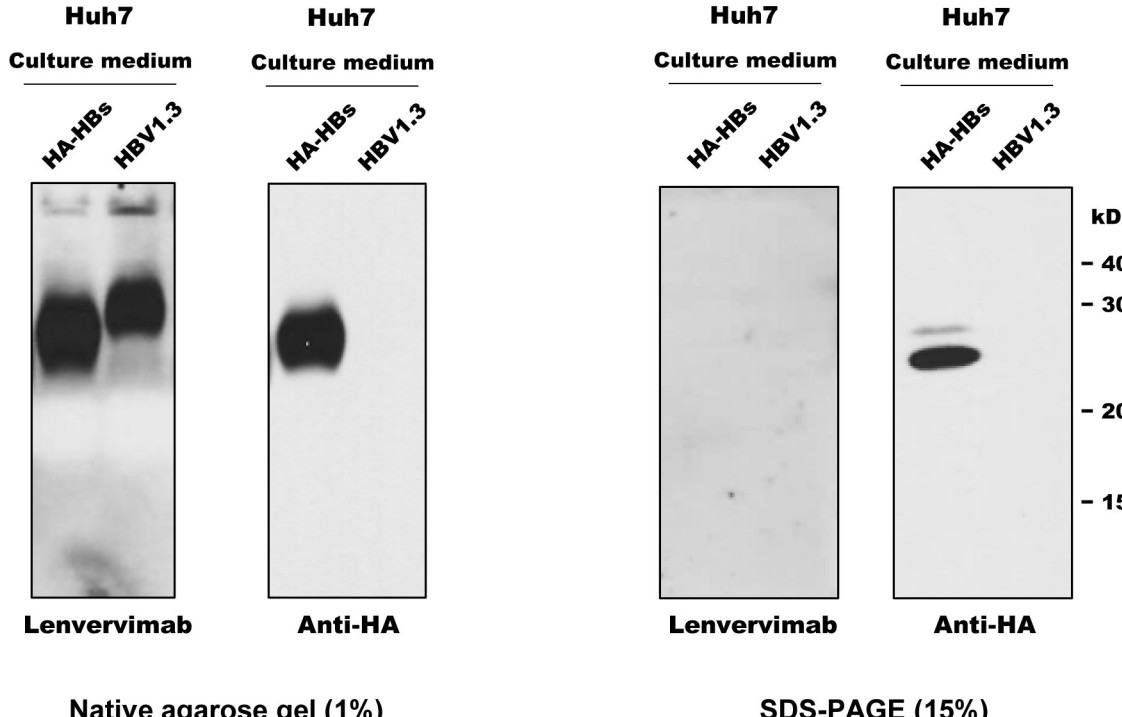

**Fig 2. Lenvervimab binds HBsAg only in nondenaturing conditions.** Aliquots of culture media from Huh7 cells transfected with pHM6-HA-HBs or HBV1.3 were separated on a 1% agarose gel in nondenaturing condition (Two panels shown on the left) or a 15% polyacrylamide gel in denaturing condition (shown on the right). Protein samples transferred to PVDF membrane were probed with Lenvervimab or anti-HA antibody, respectively.

contribution to viral replication or disease has not been thoroughly investigated. To examine whether these genotypic variations affect their affinity to antibody, we constructed 23 S clones that belong to all 10 genotypes (Table 1). (Aligned amino acid sequences are in the Supporting Information). Aliquots of transfected cell supernatant were resolved on a nondenaturing agarose gel, capillary-transferred to nitrocellulose membranes and probed with Lenvervimab. All 23 clones tested were recognized by Lenvervimab, but their signal intensity varied from 5 (Genotype F) to 147% (Genotype C) that of a genotype D clone, referred to here as the wild-type (WT) (Fig 3A). Probing the same volume of sample with anti-HA antibody or another polyclonal anti-HBsAg antibody revealed overall similar band intensity pattern to that of Lenvervimab. Specific antibody signals of each clones, after normalization with anti-HA signals, indicated that despite being a singly cloned antibody, Lenvervimab has an overall avidity comparable with a polyclonal antibody to S antigens of all genotypes distributed worldwide (Fig 3B).

## Lenvervimab binds to S antigens of a variety of clinical mutants

Conserved in all HBV strains, a dominant antigenic determinant named 'a' is in the hydrophilic and cysteine-rich loop region (aa 117–149) of S antigen (Fig 4). Exposed on the surface of HBV particles, this region contains B cell epitopes critically recognized by convalescent and vaccine-induced immunity [16, 17]. Immunological pressure gives rise to the emergence of variants with mutations within the 'a' determinant, which can result in the failure of viral neutralization and escape from detection by diagnostic reagents [18]. Several mutations in this region have been found. The most common and stable one is G145R, the postvaccination

**Table 1. S gene clones of HBV genotypes investigated in this study.**

| Clone name | Genotype | Serotype | Gene accession |
| --- | --- | --- | --- |
| A1 | A type | adw | AY738142 |
| A2 | A type | adw | DQ412135 |
| B1 | B type | adw | DQ141618 |
| B2 | B type | adw | GU815717 |
| C1 | C type | adr | V00867 |
| C2 | C type | adr | EU410082 |
| D1 | D type | ayw | JN257193 |
| D2 | D type | ayw | AY233280 |
| E1 | E type | ayw | AB091259 |
| E2 | E type | ayw | FJ692527 |
| F1 | F type | adw | FN547205 |
| F2 | F type | adw | KJ958446 |
| F3 | F type | adw | HM348689 |
| G1 | G type | adw | KC774445 |
| G2 | G type | adw | HE652720 |
| H1 | H type | adw | JQ514480 |
| H2 | H type | adw | DQ020003 |
| H3 | H type | adw | FR821991 |
| I1 | I type | ayw | FJ023664 |
| I2 | I type | adw | FJ023660 |
| J1 | J type | ayw | AB486012 |
| ayr1 | C type | ayr | JN980286 |
| WT | D type | ayw | V01460 |

immune escape mutant first found in a child born from a carrier mother [19]. This mutation was shown to abolish anti-HBsAg antibody binding to 'a' determinant [20]. To examine whether Lenvervimab binding is affected by such a mutation, we constructed 30 clones that mimic clinical variants, each harboring a single amino acid substitution at 16 sites that span the entire 'a' determinant (Table 2). HA-tagged S protein produced from each S gene clone was resolved on a nondenaturing agarose gel and immunoblotted with antibodies as described above. Unlike the genotype clones, a single amino acid change in this domain resulted in a notable mobility shift, most likely because of the net charge change at the particle surface. Lenvervimab binding intensity varied from near zero to 136% of the WT clone (Fig 5A and 5B). As described in the genotype analysis, the antibody signals of each clones were normalized with their anti-HA signals. Indeed, the normalized signal pattern indicated an overall coherent avidity of Lenvervimab to all mutants (Fig 5C). In contrast, the polyclonal anti-HBsAg antibody signal was severely reduced in several mutants within the 2nd loop of 'a' determinant, such as K141I, P142L/S, D144A/E, G145K/R, N146S and T148I. Impaired antibody binding was reported for similar mutations in this epitope region [21]. It was notable that K141E and K141I, despite their difference in expression levels (i.e., anti-HA antibody signals), both bind coherently to Lenvervimab. D144A and D144E also shared this feature. G145K and G145R, well-known immune escape mutants, both showing low expression, also showed a consistent binding to Lenvervimab. These results indicate that unlike other anti-S antibodies that heavily depend on this critical epitope, Lenvervimab binding remains largely unaffected by mutations in this region. N146S showed a comparable binding to both antibodies. We noted that this antigen was expressed in a nonglycosylated form because of the mutation at essential glycosylation site [22] (Denaturing gel immunoblot in Supporting Information). C149R was not

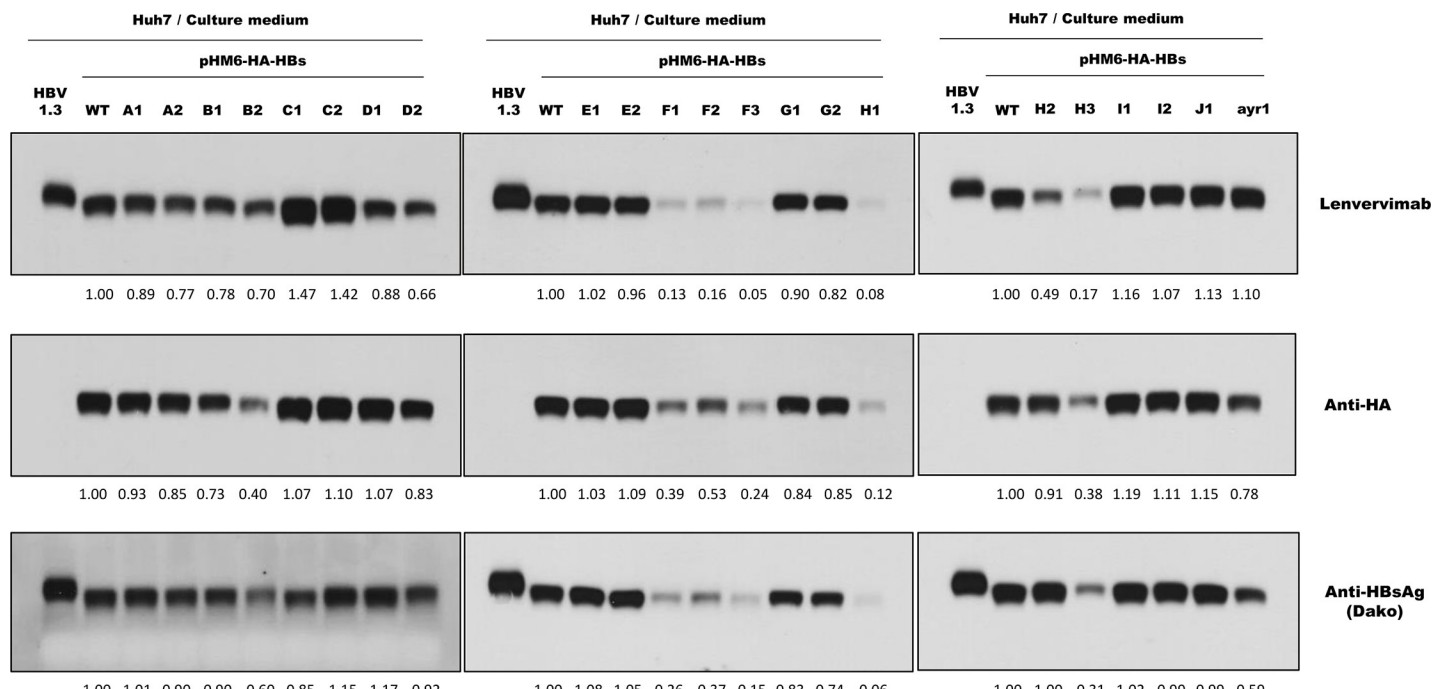

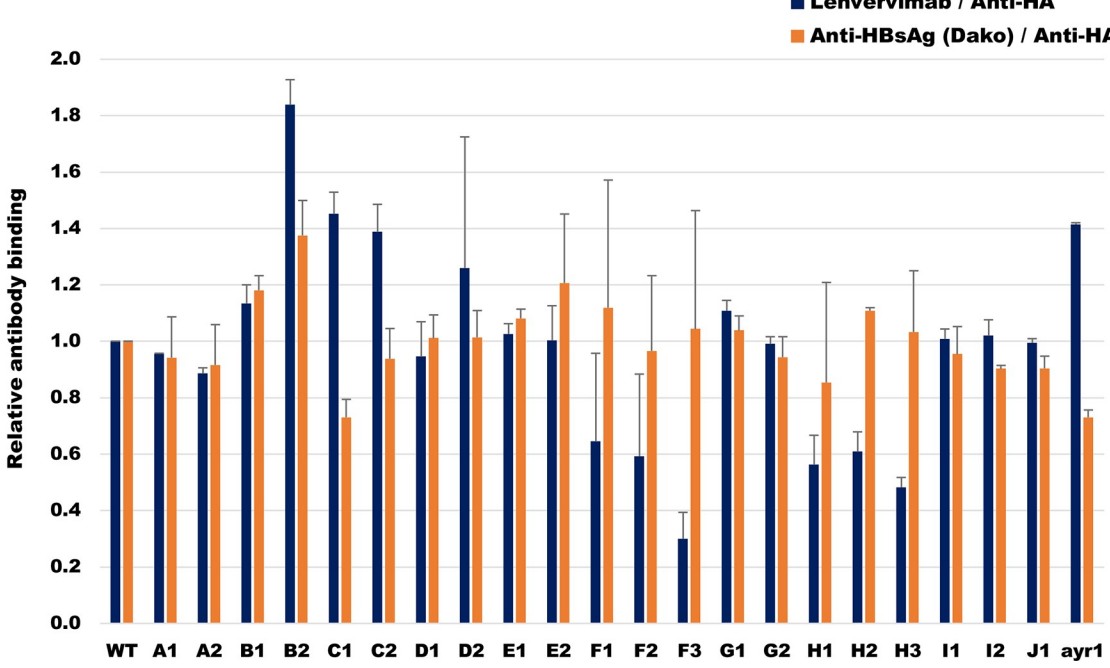

**Fig 3. Lenvervimab binds the S antigens of all viral genotypes.** (A) Aliquots of culture supernatant from transfection of various genotype S clones pHM6-HA-HBs were separated by nondenaturing agarose gels, capillary-transferred to a nitrocellulose membrane and probed with either Lenvervimab (upper), anti-HA (middle) or Dako's polyclonal anti-HBsAg (lower) antibodies. HBV1.3 was included as a control. Indicated below the image is the quantified signal intensity of each clone relative to that of a genotype D clone (ayw), referred here as wild type (WT). The image is a representative of two independent experiments. (B) Indicated as bars are signals of the Lenvervimab (in dark) and the Dako's anti-HBsAg antibody (in yellow) after normalization with anti-HA antibody signal. Average and standard errors of duplicate immunoblot analyses are shown.

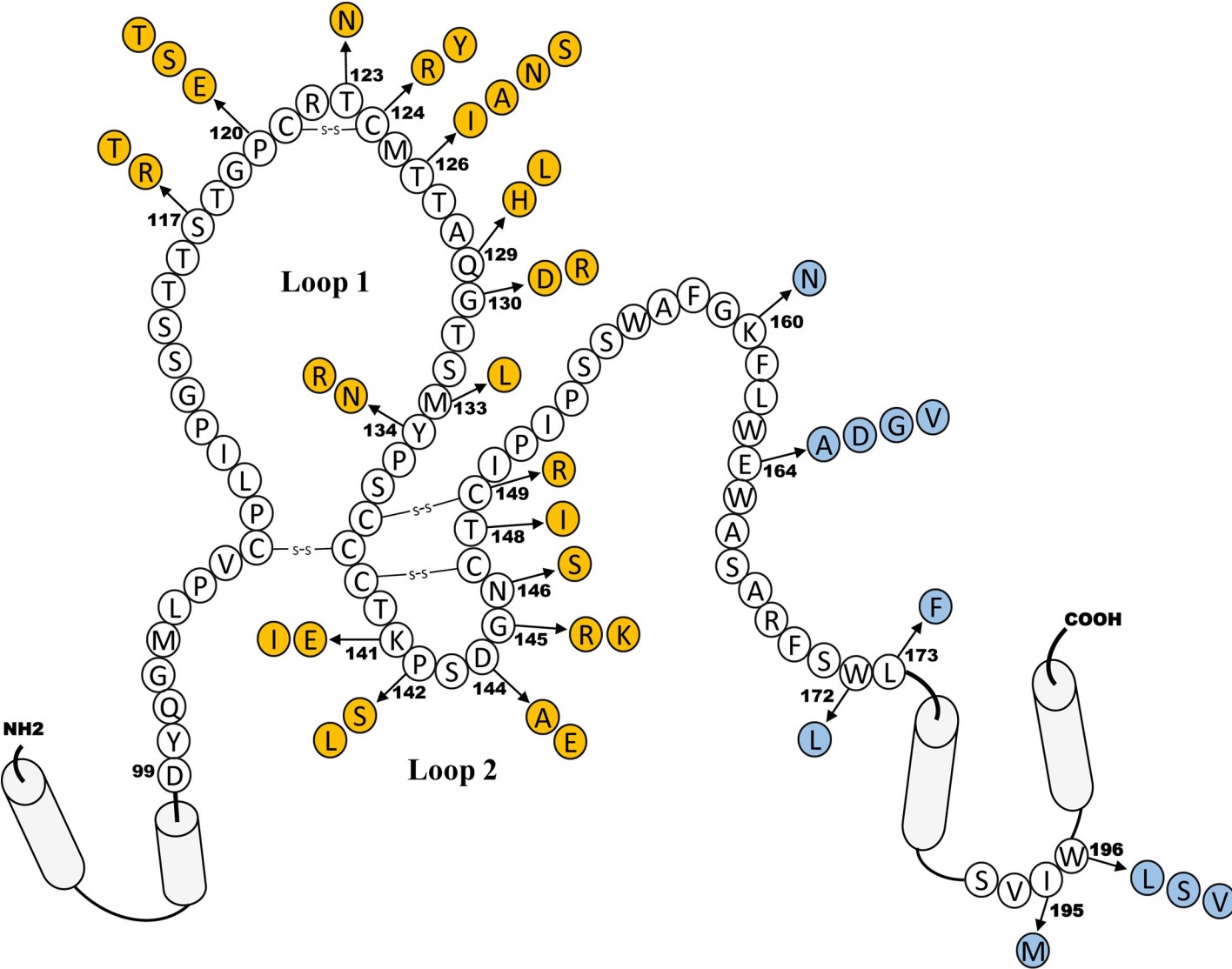

**Fig 4. HBV S gene mutations investigated in this study.** A schematic diagram of S antigen depicted with the four transmembrane domains and amino acid residues comprising the two loops of dominant antigenic region (Genotype D, ayw). In circles are mutations in the antigenic 'a' determinant (in yellow) and mutations (except K160N) in the other region of S that overlap with some viral polymerase mutations conferring resistance to nucleos(t)ide drugs.

detectable by any of the three antibodies, suggesting a critical role of this cysteine residue in the expression of S antigens.

## Lenvervimab binds S antigens of drug-resistant variants

Nucleos(t)ide analogues (NAs) are widely used as antiviral drugs for the treatment of chronic HBV infection. However, long-term therapy with NAs leads to the development of drug-resistant mutations in the viral polymerase. Because the entire S gene is embedded within the viral polymerase gene, some viral polymerase mutations that confer drug resistance can cause amino acid sequence changes in HBsAg; major ones include E164D, W172L, L173F, I195M and W196L/S/V [23, 24]. We found that these mutations, either alone or in combination of doubles, do not affect Lenvervimab binding with the exceptions of E164A/D/G/V (Fig 6). We

**Table 2. S gene mutations investigated in this study.**

| Amino acid position | Wild-type (Genotype D, ayw) | Mutant | Reference |
|---|---|---|---|
| 117 | S | R / T | [40] |
| 120 | P | E / S / T | [36] |
| 123 | T | N | [38] |
| 124 | C | R / Y | [38] |
| 126 | T | I / A / N / S | [38] |
| 129 | Q | H / L | [37] |
| 130 | G | D /R | [39] |
| 133 | M | L | [37] |
| 134 | Y | N / R | [41] |
| 141 | K | E / I | [39] |
| 142 | P | S / L | [39] |
| 144 | D | A / E | [36] |
| 145 | G | R / K | [36] |
| 146 | N | S | [41] |
| 148 | T | I | [42] |
| 149 | C | R | [39] |
| 160 | K | N | [36] |
| 164 | E | A / D / G / V | [36] |
| 172 | W | L | [36] |
| 173 | L | F | [36] |
| 195 | I | M | [36] |
| 196 | W | L / S / V | [36] |
| 172, 195 | W, I | L, M | [36] |
| 172, 196 | W, W | L, L / L,S / L,V | [36] |
| 173, 195 | L, I | F, M | [36] |
| 173, 196 | L, W | F, L / F,S / F,V | [36] |

also found that Lenvervimab does not bind K160N. The lysine residue K160 had been reported as an important antigenic element of HBsAg [25, 26]. These results indicated that the residues K160 and E164 are critical for Lenvervimab binding.

## Discussion

Human plasma fraction HBIG is currently used for newborns of chronic carrier parents as a preventive measure and for immunosuppressed patients to control viral reactivation. Considering the cost and safety issue of a biologic product from vaccinated individuals and low specific activity of HBIG, the development of more safe and efficient antibodies that can replace HBIG is still needed [27]. Lenvervimab is a recombinant IgG1-type anti-S antibody initially derived from immunoglobulin genes of a vaccinated person. Our results indicated an overall consistent avidity of Lenvervimab to S antigens of all genotypes and major subtypes distributed worldwide. We also found that Lenvervimab binding was not affected by a variety of sequence changes found in clinical variants, particularly in the antigenic 'a' determinant region. The immune escape mutant G145R is a notable one found in many cases of occult HBV infection and often undetected by routine assays, which has been attributed to the low level of circulating antigen as well as the reduced affinity to diagnostic antibodies [28]. Structural analysis on this mutant S protein indicated that a conformational change is the basis of the altered antibody binding [29]. Our results, while confirming a poor expression of this mutant antigen in

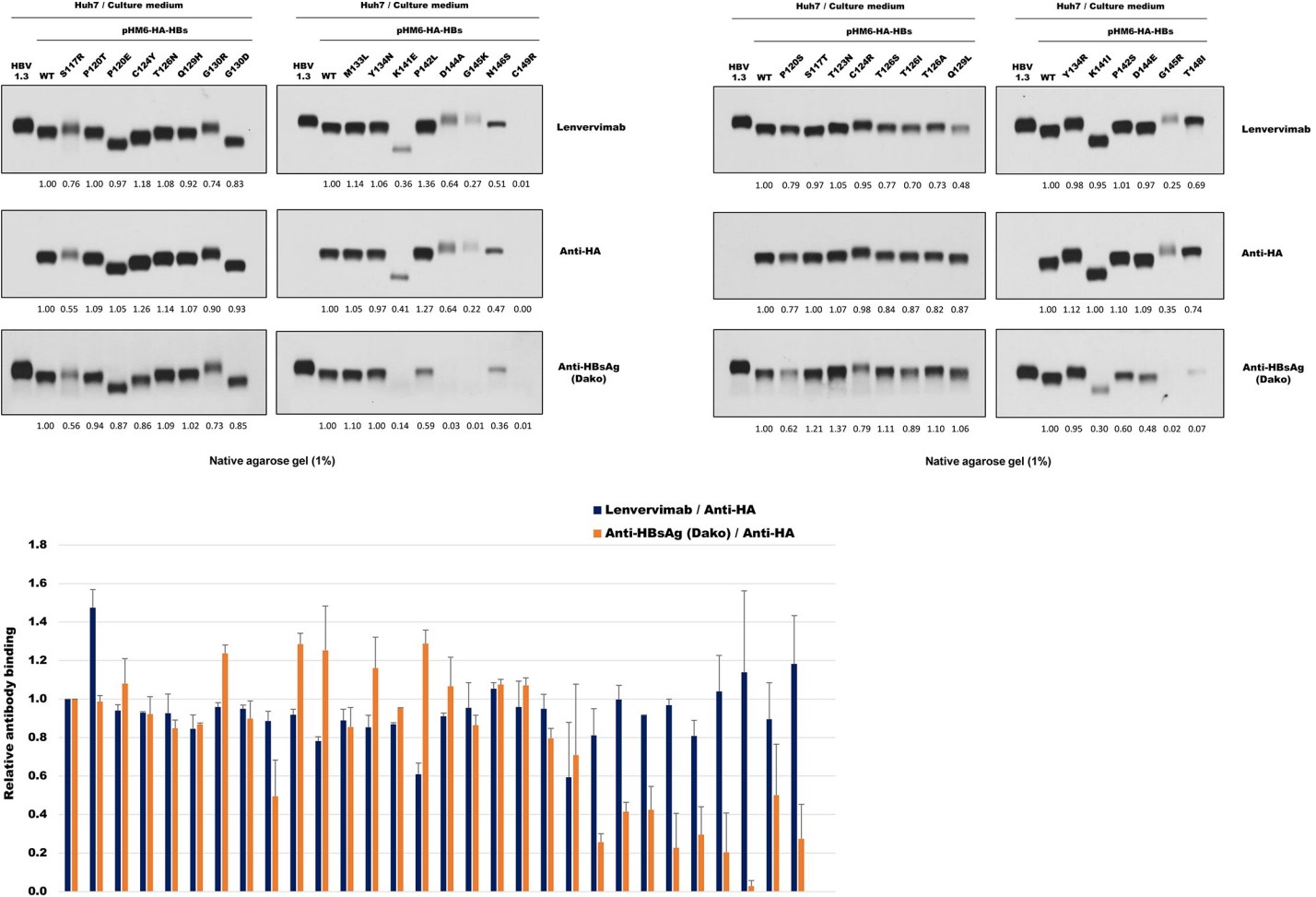

**Fig 5. Lenvervimab binds coherently to S antigens of a variety of clinical mutants.** (A, B) S antigens mimicking a variety of clinical mutants were separated by native agarose gel and immunoblotted with Lenvervimab (upper), anti-HA (middle) or polyclonal anti-HBsAg (lower) antibodies. HBV1.3 was included as a control. Indicated below the image is the quantified signal intensity of each clone relative to that of wild-type (Genotype D). (C) Bars indicate specific antibody signals normalized with the anti-HA antibody signal. Average and standard errors of duplicate immunoblot analyses are shown. Lenvervivmab (dark bars) shows consistent binding to all clones, whereas the polyclonal anti-HBsAg antibody (light bars) shows severely reduced binding to mutants at residues 141~148. No antigen was expressed from the C149R clone.

vitro, nevertheless indicate that Lenvervimab binding was not notably affected. Lenvervimab also recognized a majority of the S antigen variants that are associated with nucleos(t)ide drug-resistant mutations of the viral polymerase gene within the overlapped genome region. Our results showing Lenvervimab to have a consistent recognition of many viral genotypes and clinical variants support strongly for a preventive potential of the antibody against HBV infection.

## Materials and methods

### Cell culture, plasmids and chemical reagents

Human hepatoma Huh7 cells (JCRB0403, JCRB Cell Bank, Tokyo, Japan) and HepG2-NTCP cells [30] that express the HBV receptor protein NTCP [31] were routinely cultured at 37°C in 5% $CO_2$ in DMEM supplemented with 10% FBS (HyClone, GE Healthcare Life Science,

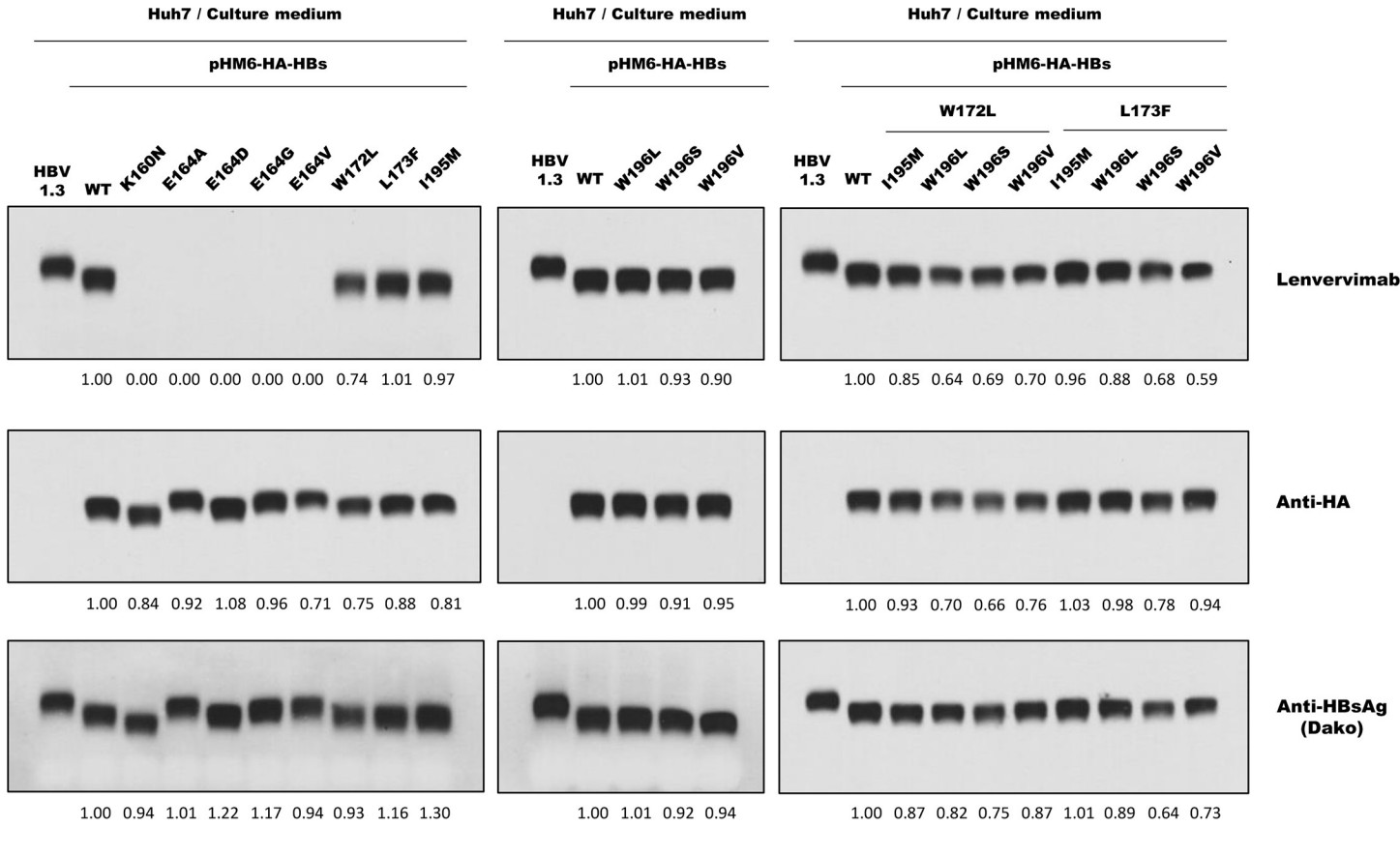

**Fig 6. Lenvervimab binds S antigens of drug-resistant variants.** (A) S antigen variations associated with drug-resistant viral polymerase mutations were separated by native agarose gel and immunoblotted with Lenvervimab (upper), anti-HA (middle) or polyclonal anti-HBsAg (lower) antibodies. Lenvervimab binds S antigens of most drug-resistant mutants except for those with mutations at residue E164. Lenvervimab binding was also impaired by mutation at residue K160, which is unrelated to drug resistance but is considered to be an antigenic element.

Chicago, IL, USA) and 50 µg/ml gentamicin (Gibco, Thermo Fisher Scientific, Waltham, MA, USA). S gene sequence was PCR amplified from the HBV (Genotype D, ayw) construct pGEM-HBV1.2 [32] and inserted into pHM6-HA (Roche, Basel, Switzerland) with a hemagglutinin (HA) epitope at the N-terminus. Viral genotype and mutant clones were generated by site-directed mutagenesis. HBV1.3-mer WT replicon [33] was used for comparative purposes. JetPEI (Polyplus-transfection, Strasbourg, France) was used for DNA transfection.

## Neutralization assay

HBV stocks were prepared from HepAD 38 cells as described [34]. HepG2-NTCP cells were incubated for 16 h at 37°C with viral inoculum (500 Genome equivalent/ml) that had been pre-incubated for 10 min at room temperature with 0.001~1 µg/ml of Lenvervimab or polyclonal anti-HBsAg rabbit antibody (NB100-62652, Novus Biologicals, Centennial, CO, USA) in DMEM containing 10% FBS, 4% PEG 8000 and 2.5% DMSO. Viral inoculum was removed after 16 h and replaced with fresh medium containing 2.5% DMSO but no PEG 8000. HBsAg in the culture media was measured after 5 days by ELISA (Genedia HBsAg ELISA 3.0, Korea Green Cross, Yongin, Korea), and intracellular core proteins were analyzed by immunoblotting with rabbit anti-HBcAg antibody as described [35].

## Immunoblotting

Viral antigens secreted in an aliquot of culture supernatant were directly loaded onto a 1% agarose gel and capillary-transferred to an Immobilon-P transfer PVDF membrane (Millipore, Burlington, MA, USA) in 20 x SSC buffer (3 M NaCl and 0.3 M sodium acetate). The membrane was probed with Lenvervimab, anti-HA antibody (SC-7392, Santa Cruz Biotechnology, Santa Cruz, CA, USA) or polyclonal goat anti-HBsAg antibody (B0560, Dako, Agilent Technologies, Santa Clara, CA, USA) in TBST buffer with 5% skim milk (BD Biosciences, San Jose, CA, USA). In some experiments, viral proteins in culture supernatants or cell lysate in RIPA buffer were separated on a denaturing polyacrylamide gel; transferred to a PVDF membrane; and probed with anti-HBcAg, anti-Hsp70 or anti-GAPDH (Santa Cruz) antibodies. HRP-conjugated secondary antibodies (Bio-Rad Laboratories, Hercules, CA, USA) and ECL reagents (Visual Protein, New Taipei City, Republic of China) were used. The signal was quantified with ImageJ software (NIH, Bethesda, MD, USA).

## ELISA

HBsAg secreted into culture supernatants was measured with Genedia HBsAg ELISA 3.0 (Korea Green Cross, Yongin, Korea) or anti-HA and HRP-conjugated secondary antibodies. Recombinant HBsAg (P4875, Abnova, Taipei City, Republic of China) was used as a standard antigen in the assay. Absorbance values at 450 nm were measured using a microplate spectrophotometer (Benchmark, Bio-Rad Laboratories, Hercules, CA, USA). Standard curve fitting and conversion to HBsAg concentration were performed by Excel software.

## Supporting information

**S1 Raw images.**
(PDF)

## Author Contributions

**Conceptualization:** Byung-Yoon Ahn, Jaesung Jung, Jung-Hwan Kim.

**Data curation:** Gi Uk Jeong, Byung-Yoon Ahn, Jaesung Jung, Ara Lee.

**Formal analysis:** Gi Uk Jeong, Woohyun Kim, Jung-Hwan Kim.

**Funding acquisition:** Jung-Hwan Kim.

**Investigation:** Gi Uk Jeong, Byung-Yoon Ahn, Jaesung Jung, Woohyun Kim, Jung-Hwan Kim.

**Methodology:** Gi Uk Jeong, Byung-Yoon Ahn, Jaesung Jung.

**Project administration:** Jaesung Jung.

**Resources:** Byung-Yoon Ahn, Hyunjin Kim, Woohyun Kim, Jung-Hwan Kim.

**Software:** Tae-Hee Kim.

**Supervision:** Byung-Yoon Ahn, Jaesung Jung, Jung-Hwan Kim.

**Validation:** Gi Uk Jeong.

**Visualization:** Kyuhyun Lee.

**Writing – original draft:** Gi Uk Jeong.

**Writing – review & editing:** Gi Uk Jeong.

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
