## [Decision Letter · Decision Letter 0]

15 Apr 2020

PONE-D-19-31577

A recombinant human immunoglobulin with coherent avidity to hepatitis B virus surface antigens of various viral genotypes and clinical mutants

PLOS ONE

Dear Dr. Ahn,

Thank you for submitting your manuscript to PLOS ONE. After careful consideration, we feel that it has merit but does not fully meet PLOS ONE’s publication criteria as it currently stands. Therefore, we invite you to submit a revised version of the manuscript that addresses the points raised during the review process.

Your manuscript was reviewed by 2 experts in the field. Both identified many important problems in your submission and produced copious comments. Please review carefully all attached comments and provide point-by-point responses.  

We would appreciate receiving your revised manuscript by May 30 2020 11:59PM. To enhance the reproducibility of your results, we recommend that if applicable you deposit your laboratory protocols in protocols.io, where a protocol can be assigned its own identifier (DOI) such that it can be cited independently in the future. For instructions see: http://journals.plos.org/plosone/s/submission-guidelines#loc-laboratory-protocols

We look forward to receiving your revised manuscript.

Kind regards,

Yury E Khudyakov, PhD

Academic Editor

PLOS ONE

2. In your Methods section, please give the sources of all cell lines used in your study.

7. Thank you for stating the following in the Acknowledgments Section of your manuscript:

"This work was supported by the GC Pharma Corp., and the Basic Science Research Program (#2018050379) of National Research Foundation of Korea. G. U. Jeong was supported by the BK21 Plus program of the Ministry of Education of Korea."

Additionally, because some of your funding information pertains to commercial funding, we ask you to provide an updated Competing Interests statement, declaring all sources of commercial funding.

In your Competing Interests statement, please confirm that your commercial funding does not alter your adherence to PLOS ONE Editorial policies and criteria by including the following statement: "This does not alter our adherence to PLOS ONE policies on sharing data and materials.” as detailed online in our guide for authors  http://journals.plos.org/plosone/s/competing-interests.  If this statement is not true and your adherence to PLOS policies on sharing data and materials is altered, please explain how.

Please include the updated Competing Interests Statement and Funding Statement in your cover letter. We will change the online submission form on your behalf.

Reviewers' comments:

Reviewer's Responses to Questions

**Comments to the Author**

1. Is the manuscript technically sound, and do the data support the conclusions?

Reviewer #1: Partly

Reviewer #2: Yes

2. Has the statistical analysis been performed appropriately and rigorously? 

Reviewer #1: N/A

Reviewer #2: Yes

3. Have the authors made all data underlying the findings in their manuscript fully available?

Reviewer #1: Yes

Reviewer #2: Yes

4. Is the manuscript presented in an intelligible fashion and written in standard English?

Reviewer #1: Yes

Reviewer #2: Yes

5. Review Comments to the Author

Reviewer #1: Reviewer :

1, The study by Gi Uk Jeong et al, investigates that a recombinant human IgG1 type anti-S antibody, named Lenvervimab, neutralizes HBV infection in a cell culture. However, this study has not used the other HBV genotypes except genotype D in infection models (HepG2-ntcp), if you want to indicate an overall consistent avidity of the antibody to the S antigens of most viral genotypes distributed worldwide, HBV genotype D is not enough in infection models.

2, Furthermore, Lenvervimab only binds nondenatured forms of HBsAg and binds the S antigens of all viral genotypes also a variety of clinical mutants, it is important to demonstrate the specific epitope Lenvervimab bind to and explain why Lenvervimab only binds nondenatured forms of HBsAg but not in denaturing conditions.

3, There is not enough evidence for Lenvervimab to show the clinical potential of the anti-S antibody for the prevention and treatment of HBV infection, maybe more experiments on the prevention and treatment of HBV infection in vivo/vitro are necessary.

4, “We noted neutralization activity at 10-fold lower concentrations of Lenvervimab when the antibody was pre-incubated with viral inoculum prior to infection (data not shown).” Why data not shown? antibody was pre-incubated with viral inoculum prior to infection maybe is the better way to neutralizes HBV infectivity.

5, Fig 1. “Lenvervimab neutralizes HBV infection in culture.” HBcAg immunoblotting is not clear. Please add the optical density information.

6, Fig2: “Lenvervimab binds HBsAg only in nondenaturing conditions” , lack of a positive control antibody.

7, The clearly functional results of Lenvervimab in this study is missing.

Reviewer #2: In the manuscript entitled ‘A recombinant human immunoglobulin with coherent avidity to hepatitis B virus surface antigens of various viral genotypes and clinical mutants’ by Jeong et al. They describe in an elegant fashion way how the monoclonal antibody ‘Lenvervimab’ interact with S antigen from different viral genotypes, mutants and clinical variants. They performed well elaborated immunoblottings and ELISA experiments demonstrating the potential clinical relevance of the Lenvimab antibody, which may in the future be a good tool or strategy for HBV infection treatments and study. It provides an original data and describes a relevant work. However, there are some issues that needs to be addressed.

Major comment

Line 61: This is one of your main results about neutralization activity. Your assay is an indirect assay for viral infectivity measurement, through HBsAg extracellular and intracellular production ratio. The ideal assay for viral neutralization assessment would be the foci or plaque neutralization assay. Anyway, my recommendation here is to not assume what has not been proved, and make clear that this is an indirect way of infectivity measurement.

Minor comments

Line 43: What do you mean by ‘assess the potential clinical utility of Lenvimab’? I understand that you may be referring to clinical treatment of infected patients, but it needs to be clarified in the phrase;

Line 46: I think a brief introduction phrase accompanied by a good reference about what is the common antigen ‘a’, would help some readers;

Line 80: Please, do not infer to your results as a ‘hoping’ that it will provide a more reliable information, trust your results and write this phrase more confident of it.

Line 94: How can you conclude that HA-HBs proteins are secreted in particle form? Please, include a reference if it is the case.

Line 137 – 140: I think this is not the focus of your work, my suggestion is to remove this phrase/speculation.

Line 167 – 170: Could you please provide reference(s) about the identification of all these clinical variants?

Line 193 – 195: You state that the denaturing gel detects only unglycosylated S protein from N146S, because this residue is responsible for glycosylation. Can you give a more detail or better explanation?

Line 223: You say that the residue E164 is important for the antigenic conformation of S antigen. However, I do not think you have data supporting that the S antigen changed its conformation. In addition, the residue E164 did not alter the ligation of the S antigen to the Anti-HBsAg. I suppose the only think you can infer here, is that this residue is critical for Lenvervimab ligation.

Line 227: Again, please explain how you can infer that K160 and E164 are important conformational antigenic determinants.

Line 254 - 256: Here, you assume that Lenvervimab neutralizes viral infectivity. However, what you observe here is partial neutralization observed by and indirect capture ELISA in cell culture system. I think you should rewrite more cautious about this conclusion.

Line 268 – 269: Please, include a reference about the relation between the clinical mutated isolates and Immune scape, occult infection and resistance to antivirals.

Line 292 – 293: Can you show the results demonstrating the antibody binding restoration with the double point mutations K160N/E164V? Otherwise, you should remove this. This is a critical criteria for Plos One, to make all data supporting your findings available.

Line 298 – 301: This is not the focus of your work. I think you should omit or reduced this statement and elaborate better about your findings.

Line 308: Do you have a reference for this receptor protein NTCP?

Line 318 – 319: What total volume did you keep your inoculum + antibodies during this 16h? Did you keep it on a rocking plate at 37°C?

Line 346: Please, elaborate better how was the statistic analyses performed

Writing/English comments

Line 39: ... recombinant monoclonal anti-S...

Line 41: ... utilizing a monkey animal model.

Line 42: In the current work, we evaluate the antibody neutralizing activity in cell culture system'

Line 44: ‘…genotypes and clinical variants. ’

Line 53: ‘… had previously been demonstrated in chimpanzee animal model. ’

Line 77: ‘...each clone variant differs in its expression... ’

Line 78 – 81: In this phrase, I think you should make more clear what is the point of adding HA glycoprotein to N-terminus of S protein clones (comparison and interaction measurement relative to Lenvervimab and AntiBsAg).

Line 134: ‘...expression/secretion of S antigen by genotypes... ’

Line 135: ‘We confirmed this result by the ELISA detection ratio between Lenvervimab and Anti-HBsAg to Anti-HA’

Line 148: ‘...referred here as wild (WT)... ’

Line 148: ‘The image is a representative of two independent experiments... ’

Line 165: ‘...and the most common and stable one is the G145R,... ’

Line 166: ‘...from/of a carrier mother. ’

Line 222: ‘…S gene mutations (Table 2) did not…’

Line 266: ‘Because this ‘a’ epitope is in a highly conformational and hydrophilic, these…’

6. PLOS authors have the option to publish the peer review history of their article (what does this mean?). If published, this will include your full peer review and any attached files.

Reviewer #1: No

Reviewer #2: Yes: Marcilio Jorge Fumagalli

---

## [Author Response · Author response to Decision Letter 0]

14 Jun 2020

Reviewer #1:

1, The study by Gi Uk Jeong et al, investigates that a recombinant human IgG1 type anti-S antibody, named Lenvervimab, neutralizes HBV infection in a cell culture. However, this study has not used the other HBV genotypes except genotype D in infection models (HepG2-ntcp), if you want to indicate an overall consistent avidity of the antibody to the S antigens of most viral genotypes distributed worldwide, HBV genotype D is not enough in infection models.

 Yes, only the genotype D virus was examined for neutralization. Central to its neutralization, however, lies the antibody binding of antigens. In this regard, we have examined Lenvervimab binding to antigens of many genotypes and clinical variants, which comprise the major portion of this work.

2, Furthermore, Lenvervimab only binds nondenatured forms of HBsAg and binds the S antigens of all viral genotypes also a variety of clinical mutants, it is important to demonstrate the specific epitope Lenvervimab bind to and explain why Lenvervimab only binds nondenatured forms of HBsAg but not in denaturing conditions.

No specific (linear) epitope could be defined for Lenvervimab. Consistent binding of many sequence variations in the structurally conserved antigenic regions, and the requirement of an exclusively nondenaturing condition strongly suggest that Lenvervimab binding more likely depends on antigen conformations.

3, There is not enough evidence for Lenvervimab to show the clinical potential of the anti-S antibody for the prevention and treatment of HBV infection, maybe more experiments on the prevention and treatment of HBV infection in vivo/vitro are necessary.

Our data showing consistent recognition of many viral genotypes and clinical variants support the potential value of Lenvervimab for preventive purposes. We have revised Abstract, Introduction and Discussion, focusing on this point.

4, “We noted neutralization activity at 10-fold lower concentrations of Lenvervimab when the antibody was pre-incubated with viral inoculum prior to infection (data not shown).” Why data not shown? antibody was pre-incubated with viral inoculum prior to infection maybe is the better way to neutralizes HBV infectivity.

We have taken the current protocol to keep the volume of viral inoculum and concentration of virus and antibody constant throughout the preincubation and infection periods. We have removed the quoted paragraph as it was not fully supported by data. 

5, Fig 1. “Lenvervimab neutralizes HBV infection in culture.” HBcAg immunoblotting is not clear. Please add the optical density information.

Quantitative information has been added in the figure 1.

6, Fig2: “Lenvervimab binds HBsAg only in nondenaturing conditions” , lack of a positive control antibody.

Lenvervimab is unique in this respect. Anti-HA antibody and another anti-HBsAg antibody (from Dako) bind the antigen in nondenaturing and in denaturing conditions. 

7, The clearly functional results of Lenvervimab in this study is missing.

Specific recognition and binding of viral antigens and many closely related variants as we have characterized are important functions of the antibody.

Reviewer #2: In the manuscript entitled ‘A recombinant human immunoglobulin with coherent avidity to hepatitis B virus surface antigens of various viral genotypes and clinical mutants’ by Jeong et al. They describe in an elegant fashion way how the monoclonal antibody ‘Lenvervimab’ interact with S antigen from different viral genotypes, mutants and clinical variants. They performed well elaborated immunoblottings and ELISA experiments demonstrating the potential clinical relevance of the Lenvervimab antibody, which may in the future be a good tool or strategy for HBV infection treatments and study. It provides an original data and describes a relevant work. However, there are some issues that needs to be addressed.

Major comment

Line 61: This is one of your main results about neutralization activity. Your assay is an indirect assay for viral infectivity measurement, through HBsAg extracellular and intracellular production ratio. The ideal assay for viral neutralization assessment would be the foci or plaque neutralization assay. Anyway, my recommendation here is to not assume what has not been proved, and make clear that this is an indirect way of infectivity measurement.

Because it is difficult to measure HBV foci or plaques in cell culture, we measured the extracellular HBsAg and intracellular HBcAg to assess the viral infectivity and inhibition thereof by antibody. We have revised the text accordingly (Line 55-58). 

Minor comments

Line 43: What do you mean by ‘assess the potential clinical utility of Lenvervimab’? I understand that you may be referring to clinical treatment of infected patients, but it needs to be clarified in the phrase;

We have revised the term to indicate ‘preventive utility’ of the antibody (Lines 40, 47-48).

Line 46: I think a brief introduction phrase accompanied by a good reference about what is the common antigen ‘a’, would help some readers;

We have added a brief mentioning of the ‘a’ epitope (Line 44-45) as a more detailed information and reference are provided in the 3rd sections of Results. 

Line 80: Please, do not infer to your results as a ‘hoping’ that it will provide a more reliable information, trust your results and write this phrase more confident of it.

Good point. We have revised the paragraph (lines 74-76) as recommended. 

Line 94: How can you conclude that HA-HBs proteins are secreted in particle form? Please, include a reference if it is the case.

We have provided references (Lines 86-88). In our part, S protein samples in cell culture supernatant can be resolved (i.e., retained) in agarose gel. Also, the antigen samples are precipitable by centrifugation at a low speed (10,000x g).

Line 137 – 140: I think this is not the focus of your work, my suggestion is to remove this phrase/speculation.

We agree with the referee’s point and removed the paragraph and placed the sequence alignment in the Supporting Information. 

Line 167 – 170: Could you please provide reference(s) about the identification of all these clinical variants?

We have included the references in Table 2.

Line 193 – 195: You state that the denaturing gel detects only unglycosylated S protein from N146S, because this residue is responsible for glycosylation. Can you give a more detail or better explanation?

Glycosylation of S protein at N146 is well established. With anti-HA antibody we detected two bands (24 and 27 kDa) from all S clones. From N146S, only the smaller band (24 kDa) was detected (Blot image in Supporting Info). We have revised the text accordingly (Lines 157-159).

Line 223: You say that the residue E164 is important for the antigenic conformation of S antigen. However, I do not think you have data supporting that the S antigen changed its conformation. In addition, the residue E164 did not alter the ligation of the S antigen to the Anti-HBsAg. I suppose the only thing you can infer here, is that this residue is critical for Lenvervimab ligation.

Line 227: Again, please explain how you can infer that K160 and E164 are important conformational antigenic determinants.

Our data show that mutation of K160 (and E164) abolishes Lenvervimab binding, but not the binding of Dako’s anti-HBsAg antibody. We do not know whether (or how) the mutation in these residues affect antigen conformation. Mobility of the mutant in nondenaturing gel appeared normal. We have revised the paragraph (Line 187-188) accordingly.

Line 254 - 256: Here, you assume that Lenvervimab neutralizes viral infectivity. However, what you observe here is partial neutralization observed by and indirect capture ELISA in cell culture system. I think you should rewrite more cautious about this conclusion.

We have removed this part of Discussion as it (“partial neutralization”) was described in Result.

Line 268 – 269: Please, include a reference about the relation between the clinical mutated isolates and Immune scape, occult infection and resistance to antivirals.

We have provided the references in Table 2.

Line 292 – 293: Can you show the results demonstrating the antibody binding restoration with the double point mutations K160N/E164V? Otherwise, you should remove this. This is critical criteria for Plos One, to make all data supporting your findings available.

We have removed this part as recommended.

Line 298 – 301: This is not the focus of your work. I think you should omit or reduced this statement and elaborate better about your findings.

We agree with the referee’s point and removed this part. 

Line 308: Do you have a reference for this receptor protein NTCP?

We have included the reference.

Line 318 – 319: What total volume did you keep your inoculum + antibodies during this 16h? Did you keep it on a rocking plate at 37°C?

We kept the liquid volume the same to keep the concentrations and ratio of virus and antibody constant throughout the 10-min preincubation (virus + antibody) at r.t. followed by the 16-hr infection at 37°C without rocking. We have elaborated the procedure in more detail (Lines 230-232).

 Line 346: Please, elaborate better how was the statistic analyses performed

It was specified in the figure legends (Lines 64-66).

Writing/English comments

Line 39: ... recombinant monoclonal anti-S...

Line 41: ... utilizing a monkey animal model.

Line 42: In the current work, we evaluate the antibody neutralizing activity in cell culture system'

Line 44: ‘…genotypes and clinical variants. ’

Line 53: ‘… had previously been demonstrated in chimpanzee animal model. ’

Line 77: ‘...each clone variant differs in its expression... ’

Line 78 – 81: In this phrase, I think you should make more clear what is the point of adding HA glycoprotein to N-terminus of S protein clones (comparison and interaction measurement relative to Lenvervimab and AntiBsAg).

Line 134: ‘...expression/secretion of S antigen by genotypes... ’

Line 135: ‘We confirmed this result by the ELISA detection ratio between Lenvervimab and Anti-HBsAg to Anti-HA’

Line 148: ‘...referred here as wild (WT)... ’

Line 148: ‘The image is a representative of two independent experiments... ’

Line 165: ‘...and the most common and stable one is the G145R,... ’

Line 166: ‘...from/of a carrier mother. ’

Line 222: ‘…S gene mutations (Table 2) did not…’

Line 266: ‘Because this ‘a’ epitope is in a highly conformational and hydrophilic, these…’

Thanks to the referee, we have revised the phrases

---

## [Decision Letter · Decision Letter 1]

14 Jul 2020

A recombinant human immunoglobulin with coherent avidity to hepatitis B virus surface antigens of various viral genotypes and clinical mutants

PONE-D-19-31577R1

Dear Dr. Ahn,

We’re pleased to inform you that your manuscript has been judged scientifically suitable for publication and will be formally accepted for publication once it meets all outstanding technical requirements.

Kind regards,

Yury E Khudyakov, PhD

Academic Editor

PLOS ONE

Additional Editor Comments (optional):

Reviewers' comments:

Reviewer's Responses to Questions

**Comments to the Author**

1. If the authors have adequately addressed your comments raised in a previous round of review and you feel that this manuscript is now acceptable for publication, you may indicate that here to bypass the “Comments to the Author” section, enter your conflict of interest statement in the “Confidential to Editor” section, and submit your "Accept" recommendation.

Reviewer #1: All comments have been addressed

Reviewer #2: All comments have been addressed

2. Is the manuscript technically sound, and do the data support the conclusions?

Reviewer #1: Yes

Reviewer #2: Yes

3. Has the statistical analysis been performed appropriately and rigorously? 

Reviewer #1: Yes

Reviewer #2: Yes

4. Have the authors made all data underlying the findings in their manuscript fully available?

Reviewer #1: Yes

Reviewer #2: Yes

5. Is the manuscript presented in an intelligible fashion and written in standard English?

Reviewer #1: Yes

Reviewer #2: Yes

6. Review Comments to the Author

Reviewer #1: (No Response)

Reviewer #2: (No Response)

7. PLOS authors have the option to publish the peer review history of their article (what does this mean?). If published, this will include your full peer review and any attached files.

Reviewer #1: No

Reviewer #2: **Yes: **Marcilio Jorge Fumagalli

---

## [Editor Report · Acceptance letter]

21 Jul 2020

PONE-D-19-31577R1 

A recombinant human immunoglobulin with coherent avidity to hepatitis B virus surface antigens of various viral genotypes and clinical mutants 

Dear Dr. Ahn:

I'm pleased to inform you that your manuscript has been deemed suitable for publication in PLOS ONE. Congratulations! Your manuscript is now with our production department. 

Kind regards, 

on behalf of

Dr. Yury E Khudyakov 

Academic Editor

PLOS ONE